# FAIR DIFFUSION SAMPLING WITHOUT DEMOGRAPHICS

## ABSTRACT

Diffusion models have transformed generative tasks. Despite their expressive power, these models are known to amplify social biases. Existing approaches attempt to address bias during training, which is computationally intensive. Recent research has shifted focus to the sampling stage, aiming to control generative distributions across demographic groups. However, this approach relies on sensitive labels to guide generation, which raises ethical, privacy, and laborious concerns due to the complexities of sensitive data collection. Another issue with the methods requiring annotation is the difficulty in enumerating all sensitive attributes, as some ethical concerns are not always perceptible to humans. Two critical problems remain to be resolved to address bias issues in diffusion models without relying on annotations: bias detection and controlled generation. In this work, rather than focusing on debiasing a certain demographic attribute, we investigate bias as it naturally arises in the wild. We propose a novel perspective that such biases are inherently embedded within pre-trained image encoders. To validate this, we systematically analyze widely used encoder backbones and characterize their biased behaviors. Building on this insight, we introduce an approach that leverages the inherent biases of pre-trained encoders to amplify bias signals, enabling hierarchical clustering to effectively identify bias in diffusion models. To control sampling, we propose a stable demographic score that improves demographic preservation, thereby encouraging a more uniform distribution across diverse demographic groups. Our method is free from annotations, offering strong flexibility and practical utility for debiasing diffusion models.

## 1 INTRODUCTION

Recent advances in diffusion models have achieved remarkable success (Ramesh et al., 2021; Team et al., 2023; Rombach et al., 2022b), significantly impacting various applications such as image generation (Podell et al., 2023; Nikankin et al., 2022), image editing (Zada et al., 2024; Yang et al., 2023), super-resolution (Moser et al., 2024; Wang et al., 2024), among many others (Cao et al., 2024; Schneuing et al., 2022; Krishnamoorthy et al., 2023). The ability of diffusion models to produce photorealistic images has profoundly transformed paradigms in image generation.

Unfortunately, bias issues remain a significant concern within diffusion models (Ning et al., 2023). Perera & Patel (2023) report that an unconditional diffusion model can amplify existing imbalances among different genders and racial groups compared to its training dataset. Such discriminatory behavior poses ethical risks and can be particularly hazardous when diffusion models are used to generate medical images for any purpose (Kazerouni et al., 2022). Unlike in classification tasks, debiasing a diffusion model presents unique challenges. Many methods developed for classification, such as training data balancing, are insufficient to address these issues.

Table 1: Comparison with fairness-aware diffusion models. △ indicates lack of explicit definition.

| | Pre-trained compatible | Intersectional bias | No training | No demographics | Unconditional compatible |
|---|---|---|---|---|---|
| H Guidance (Parihar et al., 2024) | ✓ | △ | ✗ | ✗ | ✓ |
| Att Switch (Choi et al., 2024) | ✓ | ✗ | ✓ | ✗ | ✓ |
| Latent Editing (Kwon et al., 2023) | ✓ | ✓ | ✗ | ✗ | ✓ |
| FT Fairness (Shen et al., 2024) | ✗ | ✓ | ✗ | ✗ | ✗ |
| ITI-GEN (Zhang et al., 2023) | ✓ | ✓ | ✓ | ✗ | ✗ |
| Fair Diffusion (Friedrich et al., 2023) | ✓ | △ | ✓ | ✗ | ✗ |
| Ours | ✓ | ✓ | ✓ | ✓ | ✓ |

Existing methods that address fairness issues typically involve training diffusion models with fairness regularization in the training objective (Shen et al., 2024) or by reweighting the score estimation (Kim et al., 2024). While these approaches can mitigate unfairness to some extent, they often come at a considerable cost. For reference, fine-tuning a model on the FairFace dataset requires approximately 384 GPU hours on an Nvidia A100 GPU (Shen et al., 2024).

In the era of generative models, rather than focusing on training or fine-tuning for fairness, another line of research emphasizes interventions during the sampling stage (Kwon et al., 2023; Parihar et al., 2024; Xu et al., 2025), which is considered a more efficient method for ensuring fairness in image generation. Existing methods include the use of classifiers to guide the generation process, ensuring that the distribution of generated images is balanced w.r.t. demographic groups (Parihar et al., 2024). Alternatively, Choi et al. (2024) leverage sensitive labels to steer the generation trajectory and achieve fairer outcomes. However, these methods depend on pre-specified bias information, such as gender or race annotations. Collecting such annotations at scale is not only costly but also raises substantial privacy concerns.

Another challenge lies in addressing intersectional biases. Intersectional bias arises from the combination of multiple demographic attributes, where each attribute intersection is expected to be represented equally. For instance, with two binary attributes: Gender ($G \in \{0, 1\}$) and Age ($A \in \{0, 1\}$). We expect each of the possible combinations to have probability $\frac{1}{4}$. *Attribute Switch* (Choi et al., 2024) only addresses a *single* binary attribute, thus failing to address intersectional biases involving multiple attributes. While Parihar et al. (2024) can handle multiple attributes, it balances each attribute's distribution *individually*. This does not necessarily prevent intersectional bias: for example, the model may generate disproportionately many samples for $(G = 0, A = 0)$ and $(G = 1, A = 1)$ while producing very few samples for $(G = 0, A = 1)$ and $(G = 1, A = 0)$, even though the marginal distributions for gender and age appear balanced. *Iti-gen* (Zhang et al., 2023) employs prompt tuning to mitigate intersectional biases, but it necessitates dozens of reference images for each category and is limited only to text-to-image models.

Addressing fairness at the sampling stage is both efficient and broadly applicable, as it allows publicly released pre-trained weights to be directly leveraged for debiasing. This paper poses a critical question: *Can multiple types of bias be mitigated simultaneously in diffusion models without prior knowledge of demographics*?

In this work, rather than pre-defining a target demographic attribute for debiasing, we focus on uncovering bias information in the wild. To achieve this, we first analyze the reverse problem of the generative task, which is image encoding. Image encoders such as DINO (Oquab et al., 2023) and CLIP (Radford et al., 2021) are trained on large-scale real-world images and thus inevitably acquire social biases. Prior work in fair classification has shown the biased behavior of such encoders. For evaluation purposes only, we use widely studied demographic attributes such as gender, age, and race to measure bias in pre-trained encoders. Our systematic analysis reveals that CLIP embeds substantial demographic information. The encoded demographic information is notorious for triggering fairness issues in classification, yet we turn this limitation into an opportunity by leveraging CLIP representations to uncover the demographic structure of diffusion models.

Given the unfair demographic structure, we aim to provide guidance that promotes the representation of underrepresented groups. To achieve this, inspired by classifier-free guidance, we introduce a demographic score function that is defined as a combination of the unconditional score function and the mean of the score function of the demographic clusters. This demographic score function is then applied to strengthen the representation of underrepresented groups and to mitigate stereotypes and biases in diffusion models.

To our knowledge, this work is the first in the domain of fair diffusion generation without demographics. We compare our approach to existing work, details of which are listed in Table 1. A more detailed discussion is provided in the Related work section. To summarize, we make the following contributions: (1) The method requires no annotations, prior knowledge, or reference datasets, minimizing human efforts. (2) The method is capable of mitigating multiple sources of bias, and is applicable across both conditional and unconditional diffusion models.

## 2 RELATED WORK

**Mitigating bias in the text to image diffusion models** A major research direction in debiasing diffusion models involves addressing biases in multimodal models, specifically within the text-to-

image category. Numerous studies (Yu et al., 2023; Jung et al., 2024; Berg et al., 2022; Chuang et al., 2023; Zhu et al., 2023; Orgad et al., 2023; He et al., 2024; Sahili et al., 2024) have been proposed to address fairness issues in the text-to-image models. Those can be broadly divided into two classes: debiasing text embeddings and debiasing image embeddings. In the text modality, Chuang et al. (2023) proposed a projection operator that maps biased text embeddings onto a space orthogonal to sensitive labels, aiming to mitigate textual biases. Sahili et al. (2024) implemented chain-of-thought reasoning to systematically reduce bias through guided text prompts. Additionally, Friedrich et al. (2023) demonstrated that incorporating explicit bias descriptors in text prompts facilitates balanced content generation across various demographics. On the other hand, research targeting image embeddings has explored structural adjustments to the models. Jung et al. (2024) introduced selective feature imputation to purge sensitive attributes from latent representations, thereby refining the debiasing process. Orgad et al. (2023) recommended updates to cross-attention layers to enhance bias mitigation. Furthermore, Shen et al. (2024) has promoted the use of distributional alignment losses to fine-tune models, ensuring that generated images better represent demographic diversity.

**Mitigating bias in unconditional diffusion models** Parihar et al. (2024) introduces a distribution guidance loss that steers the diffusion model towards generating a desired distribution across sensitive attributes. The model is guided to samples corresponding to certain demographic groups through sensitive attribute classifiers. This method is compatible with pre-trained weights and can simultaneously address multiple attributes. However, it necessitates training a guidance classifier within the bottleneck layer of the UNet, making the method non-agnostic to the model architecture. Choi et al. (2024) introduces a switching mechanism where a transition point in the sampling trajectory is used to mitigate bias. This method requires taking sensitive attributes as input and tailoring a denoising network to accommodate these attributes. However, it is not compatible with open-source pretrained weights, which limits its flexibility for deployment in real-world applications.

## 3 METHOD: FAIR SAMPLING WITHOUT DEMOGRAPHICS

We define bias in diffusion models as the case where the output distribution is skewed toward certain demographic groups. Formally, denote $d \in \mathcal{D}$ as a demographics group. The bias generation yields:

$$d|X \nsim \mathcal{U}(0, |\mathcal{D}|), \tag{1}$$

where $\mathcal{U}$ represents the uniform distribution over all groups. Existing work assumes that $\mathcal{D}$ is predefined and uses classifier guidance to enforce equal opportunity across generated samples. Unlike prior work, our method achieves demographic fairness without requiring demographic annotations.

To debias a diffusion model, we first identify bias signals by leveraging the image encoder within the model. Based on the detected underrepresented groups, we then propose a controllable sampling strategy that enhances their representation during generation.

### 3.1 DEMOGRAPHIC GROUPS IDENTIFICATION

**Motivation** Research on fairness in classification has uncovered that learned latent representations remain coupled with sensitive information. This issue extends to foundation models where inherent biases in backbone models further complicate classification tasks. However, it turns out to be a 'free lunch' to identify and effectively label sensitive attributes within the latent representations. **Characteristics of image encoders** We investigate the demographic information implicitly encoded within widely used image encoders. Our analysis covers models trained under different learning paradigms: supervised learning (Pre-trained ResNet (He et al., 2016), Vision Transformer (Dosovitskiy et al., 2020)), self-supervised learning (DINOv2 (Oquab et al., 2023) and DINOv3 (Siméoni et al., 2025)), and text-guided learning (CLIP (Radford et al., 2021) and OpenCLIP (Ilharco et al., 2021)). Detailed implementation specifications for the selected encoders are provided in the appendix.

Each encoder captures varying degrees of demographic information, sometimes in subtle or imperceptible ways. For evaluation, we define demographic groups spanning both sensitive attributes (e.g., gender, age, race) and non-sensitive attributes (e.g., blond hair, eyeglasses). We assess demographic representations using the CelebA (Liu et al., 2015) and FairFace (Kärkkäinen & Joo, 2019). Specifically, for CelebA, we focus on male, young, and blond hair attributes. We evaluate race (6 classes) as the ground-truth demographic label on FairFace.

**Demographic information extraction** To examine the demographic information captured by image encoders, we apply clustering algorithms from three categories: clustering on raw latent features, dimension reduction followed by clustering, and hierarchical clustering. Detailed descriptions of the

clustering procedures are provided in the appendix. For each encoder, we extract latent representations and perform clustering.

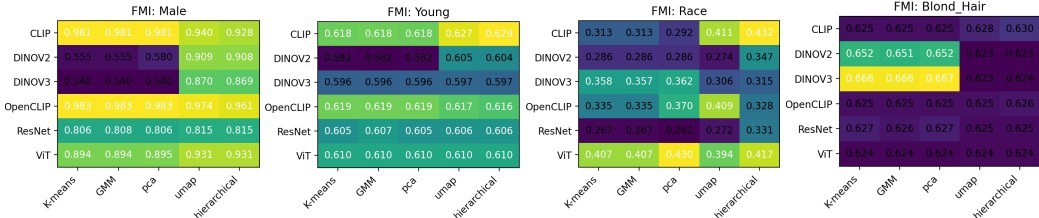

Figure 1: Confusion matrix of image encoders VS demographic group results, with values representing the Fowlkes–Mallows Index. Higher scores indicate stronger alignment with ground-truth labels. For race, since it is a multiclass attribute, the FMI is numerically lower than for binary attributes, but it still demonstrates distinguishable power. Further discussion is provided in the Appendix B.

To evaluate alignment between clustering results and ground-truth demographic labels, we use the Fowlkes–Mallows index (FMI), which quantifies consistency between clustering assignments and true labels. The FMI ranges from 0 to 1, with higher values indicating stronger agreement. A detailed introduction to the FMI is provided in the appendix. Results are shown in Figure 1, where larger values indicate higher accuracy in capturing demographic information. Our key observations are as follows:

(1) Pre-trained image encoders consistently encode sensitive attributes, with gender being the most vulnerable. Among all evaluated encoders, CLIP models exhibit the strongest discriminative power in identifying gender.

(2) Different attributes are best captured by different backbones. For example, blond hair is most effectively extracted using DINOv3, while age and gender are better captured by CLIP.

**Intersectional bias** A key challenge arises when handling multiple demographic attributes. Our findings indicate that simply increasing the number of clusters is insufficient for accurately identifying multiple demographics simultaneously. We attribute these issues to the hierarchical structuring of attributes within the representation. To verify this, we utilized the CelebA dataset, focusing on two attributes: "Male" and "Young". We extracted the representations using CLIP-ViT/L and employed UMAP to reduce the dimensionality to two. The results are visualized on the left side of Figure 2. The figure illustrates that gender information predominates at the top of the hierarchy, followed by age. Drawing on the hierarchical structure within the representation, we implement a two-stage hierarchical clustering approach to identify demographic levels at multiple scales. From Figure 1, we observe that hierarchical clustering outperforms other clustering methods in identifying sensitive attributes. We now present our hierarchical clustering algorithm.

*Coarse Clustering*: We start with using an image backbone to extract representations $\mathbf{z} = f(\mathbf{x})$, where $f$ is an image backbone, $\mathbf{z}$ denotes the latent representation. Consistent with the findings of Sohoni et al. (2020), we employ UMAP for dimensionality reduction of the latent representation. This is followed by applying the $k$-means algorithm for coarse clustering to form $c$ clusters. Throughout our experiments, we set $c = 2$, as the first level of hierarchical clustering empirically tends to capture gender related variation.

*Fine-grid Clustering:* For each coarse cluster in $\mathbf{C}$, we refine its components utilizing an over-clustering technique. For samples within each $c_i \in \mathbf{C}$, we further subdivide into $v$ subgroups, denoted as $c_{i1}, \ldots, c_{iv}$. $v$ are hyperparameters, the effect of which is analyzed in ablation study. We obtain a total of $|\mathbf{C}| = c \times v$ clusters, with each cluster assigned a distinct demographic label.

We validate the procedure by setting both the number of coarse clusters and fine-grid clusters to 2. We apply the algorithm to generate pseudo-labeled demographics, shown on the right side of Figure 2. The similarity between the left (ground truth) and right (pseudo-labeled) demonstrates strong alignment, highlighting the method's capability to accurately identify intersectional demographics.

### 3.2 DEMOGRAPHIC SCORE FUNCTION FOR AUTO-BALANCING

The previous section introduced methods for identifying demographic groups. In this section, we propose a strategy to mitigate underrepresentation through controlled sampling. Before introducing our approach, we first review commonly used controlled sampling techniques. Denote the score func-

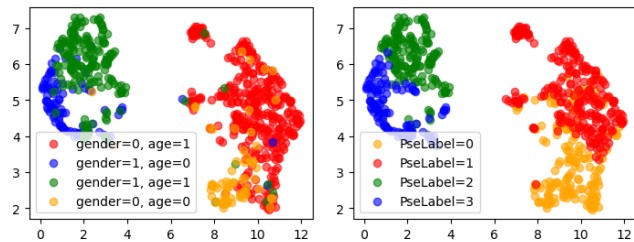

Figure 2: Intersectional demographics identification in CelebA. *Left:* Ground truth. *Right:* Pseudo-labeled demographic group generated by hierarchical algorithm.

tion as $\mathbf{s}_\theta(\mathbf{x}(t), t) \approx \nabla_{\mathbf{x}(t)} \log p(\mathbf{x}(t))$. Ho & Salimans (2022) introduced classifier-free guidance to control generation under a given condition. Specifically, the conditional score function can be approximated as

$$\nabla_{\mathbf{x}(t)} \log p(\mathbf{x}(t)|c) \approx (1+\omega)\mathbf{s}_\theta(\mathbf{x}(t), t, c) - \omega\mathbf{s}_\theta(\mathbf{x}(t), t, \varnothing), \qquad (2)$$

where $\mathbf{s}_\theta(\mathbf{x}(t), t, c)$ denotes the conditional score function with respect to condition $c$, $\omega$ is the controlled strength.

Note that a common approach to incorporate a condition into the score function $\mathbf{s}_\theta(\mathbf{x}(t), t, c)$ is through cross-attention in a U-Net architecture, which entails modifying the backbone model and requires retraining (Rombach et al., 2022b). In the context of fair generation, Choi et al. (2024) proposes a U-Net modification that incorporates sensitive attributes as input to enable conditional generation. In this section, we introduce a plug-and-play method that achieves fairness without requiring any retraining of the generative model.

**Demographic score guided fair generation.** The objective of a fair generation is to align the generative sample distribution with a uniform distribution over demographics, as introduced in Section 3. To enforce uniformity, we allocate $\frac{N}{|\mathbf{D}|}$, samples to each demographic cluster identified in Section 3.1, where $N$ denotes the total number of generated samples. Consequently, the problem reduces to controlling the generation process to satisfy a given demographic attribute $d \in \mathcal{D}$.

To guide generation according to a demographic attribute, we define the demographic representative score function as

$$\mathbf{s}_\theta(\mathbf{x}(t), t|d) = \mathbb{E}_{\mathbf{x}(t)}[\mathbf{s}_\theta(\mathbf{x}(t), t|S(\mathbf{x}(T)) = d)], \qquad (3)$$

where $S(\cdot)$ indicates the clustering algorithm introduced in Section 3.1. The computation of the demographic representative score function requires no modification to the underlying noise prediction architecture (e.g., U-Net, Diffusion Transformer). It aggregates samples within each demographic cluster and averages their score functions. Directly applying this representative score function leads to samples collapse around the mean of each demographic cluster, thereby reducing intra-cluster diversity. As illustrated in Figure 3, using only the demographic representative score function in the generation process produces nearly identical samples within the same cluster.

To address this issue, we incorporate the unconditional score function as a prior to preserve diversity.

$$\mathbf{s}_\theta(\mathbf{x}(t), t, d) = \lambda \mathbf{s}_\theta(\mathbf{x}(t), t|d) + (1-\lambda)\mathbf{s}_\theta(\mathbf{x}(t), t), \qquad (4)$$

where $\lambda \in [0, 1)$ is a hyperparameter that balances demographic consistency and generation diversity. equation 4 defines the demographic score function, which combines two components. The first term is the demographic representative score, guiding the sample toward the corresponding demographic cluster. The second term is the unconditional score function, which drives the sample toward high-density regions of the image distribution. A simple combination of the two score functions can trigger stability issues: when the directions of two score functions are highly misaligned, the effective denoising step size toward the high-density region may be unintentionally reduced, which results in degraded image quality or even generation failure. To address this, we orthogonalize the demographic representative score function:

$$\mathbf{s}_\theta^\perp(\mathbf{x}(t), t, d) = \mathbf{s}_\theta(\mathbf{x}(t), t|d) - \frac{\langle \mathbf{s}_\theta(\mathbf{x}(t), t|d), \mathbf{s}_\theta(\mathbf{x}(t), t) \rangle}{\langle \mathbf{s}_\theta(\mathbf{x}(t), t), \mathbf{s}_\theta(\mathbf{x}(t), t) \rangle} \mathbf{s}_\theta(\mathbf{x}(t), t). \qquad (5)$$

The stable demographic score function is computed by:

$$\mathbf{s}_\theta(\mathbf{x}(t), t, d) = \lambda \mathbf{s}_\theta^\perp(\mathbf{x}(t), t, d) + \mathbf{s}_\theta(\mathbf{x}(t), t). \qquad (6)$$

Figure 3: Unconditional image generation results under different sampling strategies are shown for DDIM, equation 3, and equation 6. Since DDIM involves no guidance, it produces the highest diversity, but the generations lack within-cluster consistency and exhibit significant gender bias. Using the demographic representative score function equation 3 enforces strong within-cluster consistency, but collapses diversity, which is undesirable. Our proposed method equation 6 achieves both: it preserves demographic attributes within each cluster while simultaneously maintaining diversity.

In equation 6, the first term guides the generation trajectory toward the desired demographic cluster, emphasizing demographic preservation, while the second term focuses on image quality and maintains diversity within each demographic group. We ablate generation results under three settings: (1) vanilla DDIM, (2) the demographic representative score function (equation 3), and (3) the combined demographic score function (equation 6). All methods are applied using the same pretrained weights from CelebA-HQ-256. For evaluation, we partition the data into four demographic clusters and generate samples from each cluster. From Figure 3, we observe that vanilla DDIM produces images without demographic consistency across clusters. Using equation 3 enforces strong demographic consistency, but this comes with reduced diversity, which is undesirable for fair generation. In contrast, our proposed demographic score function achieves both: it ensures demographic consistency while simultaneously preserving diversity within each cluster.

**Analysis.** We analyze the distribution of samples produced by our sampling scheme. Theorem 1 quantifies its effects on (1) within-cluster consistency and (2) within-cluster diversity of the generated sample.

*Setting.* Fix $d \in \mathcal{D}$. When $t \to T$, the sampling distribution generated by equation 4 converges to a density of the form:

$$p_\lambda(x|d) = \frac{p(x|d)^\lambda p(x)^{1-\lambda}}{Z(\lambda)}, \quad Z(\lambda) = \int p(x|d)^\lambda p(x)^{1-\lambda} \mathrm{d}x, \tag{7}$$

where $p(x)$ denotes the unconditional density and $Z(\lambda)$ is the normalizing constant. Define the log-likelihood ratio $l(x) = \log \frac{p(x|d)}{p(x)}$. Theorem 1 establishes that, under the proposed score function, the resulting distribution exhibits within-cluster demographic consistency while maintaining diversity within each demographic cluster.

**Theorem 1** (Within-cluster consistency and sample diversity)**.** *Given the demographic score function* $\mathbf{s}_\theta(\mathbf{x}(t), t, d)$, *after convergence:*

$$\frac{d}{d\lambda} KL(p_\lambda(x|d)||p(x|d)) = -(1-\lambda) Var(l(x)) < 0.$$

*Moreover, the within-cluster entropy:*

$$H(p_\lambda(x|d)) \geq H(p(x|d)),$$

*where* $H(\cdot)$ *denotes the entropy.*

**Takeaway** The theorem shows that incorporating the demographic score yields a monotonic improvement in within-cluster consistency, as indicated by the decreasing KL divergence with respect to the conditional prior of the demographic attribute. At the same time, it increases sample diversity within each cluster, with the entropy guaranteed to be no lower than that of the conditional prior.

## 4 EXPERIMENTS

As a debiasing framework, our proposed method can be broadly applied to both unconditional and conditional generation tasks. In this section, we design experiments to address the following research questions: **RQ1**: How does our method perform in unconditional diffusion models, where debiasing approaches targeting text modalities fail? **RQ2**: How does our method perform in conditional

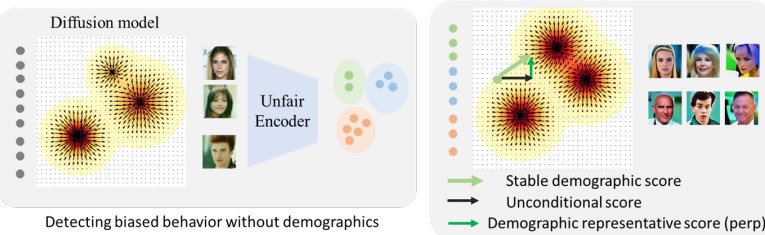

Figure 4: Fair sampling pipeline: In the first step, we generate samples from the diffusion model and pass them through an unbiased image encoder to detect biases. In the second step, we apply the stable demographic score to adjust the sampling process, ensuring that the generated samples are evenly distributed across the identified demographic clusters.

diffusion models for mitigating stereotype issues? **RQ3**: How do generated images using our method contribute to downstream tasks such as unsupervised representation learning? **RQ4**: In Section 3.1, we investigated the demographic information encoded by different image backbones. How does the choice of backbone affect fairness performance?

### 4.1 DEBIASING UNCONDITIONAL MODEL

**Implementation** Following the existing protocols for evaluating unconditional generation tasks (Parihar et al., 2024; Choi et al., 2024), we utilized unconditional diffusion models trained on CelebA-HQ from Huggingface. We employed CLIP ViT-H as the backbone to extract latent representations. For demographic group identification, we applied hierarchical clustering with dimensionality reduction UMAP, selecting $c = 2$ and $v = 3$, which yielded 6 clusters. During the fair sampling stage, we set the guidance strength to $\lambda = 0.07$. Detailed implementation specifications are provided in the Appendix E, and code is available in the supplementary materials.

**Comparison methods** We benchmark our approach against state-of-the-art fair sampling methods that use annotations, including H-distribution guidance (Parihar et al., 2024), Latent-based editing (Kwon et al., 2023), and Attribution switching (Choi et al., 2024), with a detailed discussion of these methods provided in the Appendix H. To ensure a fair comparison, we provide appropriate annotations for each method. We denote {G, A, R} as guidance for Gender, Age, and Race, respectively.

Table 2: Results of the unconditional diffusion model on CelebA dataset

| Method | Annotation | Gender ↓ | Age ↓ | Gender × Age ↓ | Race ↓ | Avg ↓ | FID ↓ |
|---|---|---|---|---|---|---|---|
| H Guidance | {G,A,R} | 22.06 | 48.69 | 49.36 | 55.91 | 44.00 | 80.72 |
| Latent Editing | {G,A,R} | 5.23 | 35.74 | **24.06** | 30.68 | **23.93** | 74.11 |
| H Guidance | {G} | 16.97 | 58.55 | 46.48 | 58.77 | 45.19 | 78.49 |
| Latent Editing | {G} | **0.56** | 39.16 | 27.40 | 62.47 | 32.40 | 62.96 |
| Att Switch | {G} | 6.79 | 52.32 | 41.29 | 71.46 | 42.96 | 103.68 |
| H Guidance | {A} | 23.48 | 43.99 | 49.33 | 59.73 | 44.13 | 77.57 |
| Latent Editing | {A} | 11.31 | **17.54** | 29.36 | 56.62 | 28.71 | 84.04 |
| Att Switch | {A} | 12.73 | 38.18 | 41.79 | 63.60 | 39.07 | 147.85 |
| H Guidance | {R} | 23.05 | 59.54 | 49.54 | 50.22 | 45.59 | 78.06 |
| Latent Editing | {R} | 17.38 | 23.00 | 27.26 | **30.35** | 24.50 | 81.56 |
| Random Sampling(DDIM) | *None* | 31.76 | 50.67 | 51.98 | 60.45 | 48.72 | 71.50 |
| Ours | *None* | 0.71 | 43.84 | 36.10 | 42.94 | 30.90 | 78.11 |

**Unconditional generation evaluation** We employ an accurate and robust pseudo-labeling network to generate surrogate-sensitive attributes for evaluation purposes. Concretely, we utilize the DINO-V2 (Oquab et al., 2023) backbone and train a linear classifier to assign demographic labels. For a comprehensive evaluation, we consider the following settings: *binary attributes*, *intersectional attributes*, and *multi-class attributes*. For binary attributes, we examine age and gender. Intersectional attributes include all combinations of two binary attributes. Multi-class attributes contain race: {Asian, White, Black, Middle Eastern, Latino/Hispanic, Indian}. We employ the Fairness Discrepancy (FD) metric defined as $100 \times ||\hat{p} - \mathbb{E}(\hat{a})||$, where $\hat{p}$ is a uniform vector matching the dimensions of $\hat{a}$, and $\hat{a}$ represents the sensitive labels predicted by a high-accuracy classifier (Parihar et al., 2024). The overall image quality is measured using the Fréchet Inception Distance (FID) score.

**Result** We present generation results on Tables 2. Latent Editing achieves the fairest outcomes. It should be noted that Latent Editing is not inherently a fairness-aware generation method. To adapt it for fairness, for example, in balancing gender attributes, the method is modified by first generating 50% male and 50% female images using Latent Editing, and then combining these images for evaluation. This modification naturally favors balanced presentation. However, it is clearly observable that the edited images exhibit poor performance in terms of quality, a finding that is

consistent with those reported in (Parihar et al., 2024). Attribute Switch necessitates a tailored UNet architecture to take annotations and requires training from scratch to implement its algorithm, making it the least flexible and efficient method evaluated. It takes binary input and is thus unable to debias race attributions. The method reduces bias effectively when specific attributes are included. However, it inadvertently increases bias when those attributes are not considered. For instance, when addressing gender bias, it unintentionally exacerbates racial disparities, which is not a desirable outcome even with annotations. Our proposed method achieves a balance between fairness and generation quality. Importantly, it significantly reduces intersectional bias even without any annotations. In terms of image quality, the sampling method produces samples comparable to those from Random sampling, avoiding strong editing artifacts and preserving visual authenticity.

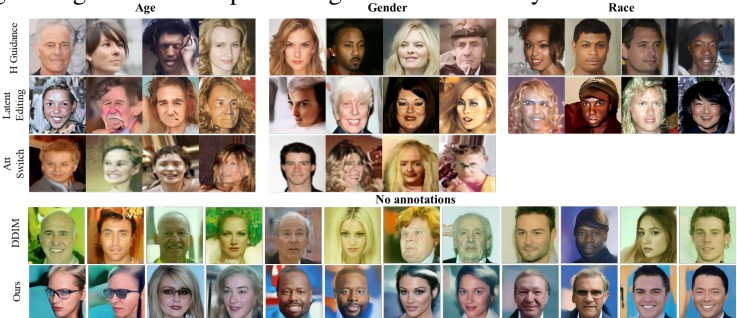

Figure 5: Visualization of image generation quality using fairness-aware methods on CelebA dataset.

## 4.2 ANTI-STEREOTYPE IN TEXT-TO-IMAGE MODELS

We present our method to mitigate stereotypes inherent even in the state-of-the-art Text-to-Image models. We test on two generation frameworks, *Stable Diffusion V2.1* (SDV2.1) (Rombach et al., 2022a) and *Stable Diffusion V3* (SDV3) (Esser et al., 2024), representing diffusion models with UNet and Transformer architectures, respectively. We focus on mitigating stereotypes, where sensitive attributes are often associated with occupations or appearances. To evaluate this, we use the prompts Doctor, Manager, Firefighter, and Bald, generating 200 images per prompt for each model.

**Evaluation** Using a supervised classifier trained on a dataset like CelebA for gender classification on a different dataset such as stable diffusion images can lead to distribution shift issues and unreliable results. To avoid this, we employ zero-shot classification using CLIP-ViT-H/14. The accuracy of this classifier is verified on annotated datasets. Further details are provided in the Appendix G.

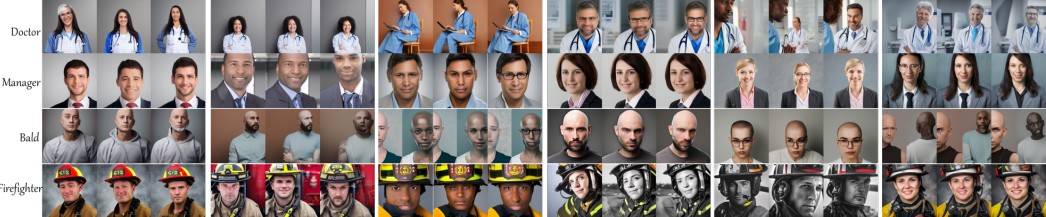

Figure 6: Conditional generation results using SDV2.1. Three images are randomly selected from each cluster. Zoom in to check image quality.

**Results** From Table 3, we observe that both SDV2.1 and SDV3 exhibit strong stereotypical biases. After applying our method, the fairness discrepancy is reduced significantly. For image generation quality, we refer to Figures 6 and 9, which show that our method achieves high image quality. We note a slight drop in race fairness evaluation for SDV2.1. We hypothesize two possible reasons: (1) some generated images may contain multiple people, which could affect the evaluation (See row 3, Figure 6), and (2) our method may prioritize other attributes, as they are more easily distinguishable, which could potentially reduce racial diversity.

## 4.3 APPLICATION FOR CLASSIFICATION TASKS

Beyond the generation task, we investigate the effect of using generative images on downstream tasks. The bias in classification tasks often stems from data skewness. Existing studies have shown that a more balanced dataset can reduce bias in classification models Idrissi et al. (2022). We extend this investigation to assess the utility of fairly generated samples for classification tasks. To validate this,

Table 3: Evaluation results on SDV2 and SDV3.

| | | Doctor | | Manager | | Bald | | Firefighter | | Avg | |
| | | Baseline | Ours | Baseline | Ours | Baseline | Ours | Baseline | Ours | Baseline | Ours |
|---|---|---|---|---|---|---|---|---|---|---|---|
| SDV2.1 | G $\downarrow$ | 44.19 | 22.63 | 55.86 | 38.18 | 41.15 | 28.99 | 64.33 | 25.46 | 51.38 | 28.81 (43.93%) |
| | A $\downarrow$ | 37.89 | 37.47 | 36.06 | 25.46 | 65.43 | 50.20 | 24.99 | 9.90 | 41.09 | 30.76 (25.14%) |
| | G $\times$ A $\downarrow$ | 51.41 | 26.92 | 56.78 | 33.26 | 63.09 | 44.37 | 52.99 | 27.76 | 56.07 | 33.08 (41.00%) |
| | R $\downarrow$ | 47.59 | 40.76 | 48.42 | 43.79 | 55.32 | 67.88 | 48.21 | 55.39 | 49.89 | 51.96 (4.15%) |
| SDV3 | G $\downarrow$ | 48.08 | 29.90 | 53.03 | 0.35 | 70.71 | 62.27 | 67.18 | 29.90 | 59.75 | 30.61 (48.77%) |
| | A $\downarrow$ | 24.75 | 21.46 | 19.80 | 2.46 | 58.69 | 55.23 | 45.25 | 29.20 | 37.12 | 27.09 (27.02%) |
| | G $\times$ A $\downarrow$ | 44.88 | 27.86 | 45.83 | 25.62 | 77.64 | 55.25 | 66.96 | 53.86 | 58.83 | 40.65 (30.90%) |
| | R $\downarrow$ | 36.85 | 35.27 | 59.49 | 58.65 | 83.33 | 65.54 | 67.70 | 49.81 | 61.84 | 52.32 (15.39%) |

we generate images using an unconditional diffusion model trained on the CelebA dataset. These images are generated separately by the different baseline methods and our proposed methods. Since these generated images have no labels, we employ the self-supervised learning method SimCLR Chen et al. (2020) to extract feature representations. We use the SimCLR method to train a ResNet-18 backbones with different image set generated by different methods. We then freeze the SimCLR backbone network and attach a linear classifier to conduct the classification task through a linear probe. The linear classifier is trained on the CelebA training set with the Blond Hair annotation, optimized with an Empirical Risk Minimization objective.

For testing, we utilize the CelebA test set to perform the classification, evaluating the results based on both utility and fairness. In our fairness assessment, we consider 'male' as the sensitive attribute, a common practice in fairness studies. For a comprehensive fairness evaluation, we measure demographic parity (DP), equal opportunity (EOp), and equalized odds (EOd) (Lu et al., 2024). The results are presented in table 4. It is evident that after implementing our method, there is a significant improvement in fairness within the classification tasks, approaching the best performance achieved by H-Guidance with gender guidance. These results indicate that our method substantially benefits fairness in downstream classification tasks.

Table 4: Linear probe: $y =$ 'blond hair' and $a =$ 'male'.

| Methods | Annotation | DP $\downarrow$ | EOp $\downarrow$ | EOd $\downarrow$ | ACC $\uparrow$ |
|---|---|---|---|---|---|
| CelebA-subset | None | 17.30 | 59.60 | 31.44 | 94.13 |
| Random sampling | None | 14.29 | 42.68 | 22.29 | 91.87 |
| Att Switch | {G} | 8.92 | 34.09 | 17.45 | 91.05 |
| H-Guidance | {G} | 9.26 | 32.86 | 16.78 | 91.30 |
| Latent-editing | {G} | 10.92 | 39.62 | 20.42 | 91.79 |
| Ours | None | 10.00 | 36.50 | 19.31 | 90.40 |

Table 5: Ablation Study on the backbone selection.

| Backbone | Gender $\downarrow$ | Age $\downarrow$ | Blond Hair $\downarrow$ |
|---|---|---|---|
| CLIP | **0.71** | **43.84** | 36.10 |
| DINO V3 | 30.41 | 46.57 | **19.79** |
| DINO V2 | 2.12 | 45.96 | 22.43 |

## 4.4 ABLATION STUDY

**Image encoders.** In Section 3.1, we analyzed how different image encoders capture demographic information and observed that CLIP models embed the most sensitive features. In this section, we conduct an ablation study by changing different image encoders to evaluate fairness outcomes. We consider DINO V3 and DINO V2 as image backbones. We examine the distribution of generated images with respect to Gender, Race, and Blond Hair. A more balanced attribute distribution in the generated samples indicates that the corresponding attribute is more strongly encoded, making it more distinguishable in the latent representation. The results are presented in Table 5. Compared to the CLIP model, DINO V3 is less effective at balancing gender and age, but achieves stronger balance on the blond hair attribute, which is consistent with the findings in Section 3.1. These results further validate that CLIP models are more suited for debiasing sensitive attributes. Additional ablation results are presented in Appendix I.

## 5 CONCLUSION

In this work, we address the challenge of mitigating bias in diffusion models without demographics. Our method is not to target specific biased attributes, but rather to pursue a more general approach aimed at debiasing in the wild. We systematically analyze the behavior of image encoders and find that CLIP demonstrates a strong capability to capture sensitive demographic information. To promote a fairer generation, we employ CLIP to identify demographic groups and design a stable demographic score that guides the generation process, achieving both fairness awareness and intra-group diversity. Our experiments on both unconditional and conditional text-to-image diffusion models reveal that the proposed method markedly improves fairness, without relying on annotations.

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

## A  DETAILS OF SECTION 3.1

**Implementation of Image encoders**  We evaluate a collection of widely used modern image encoders with strong performance. Specifically, we consider pre-training paradigms including supervised learning, self-supervised learning, and text-guided training. From a structural perspective, we examine both convolutional neural networks (CNN) and vision transformers (ViT).

For supervised pre-training, we adopt ResNet-50 (He et al., 2016) and ViT-L/16 (Dosovitskiy et al., 2020) trained on ImageNet-21k. For self-supervised learning, we include DINOv2 (ViT-L) (Oquab et al., 2023) and DINOv3 (7B) (Siméoni et al., 2025). For text-guided encoders, we consider CLIP (ViT-L, OpenAI) (Radford et al., 2021) and OpenCLIP (ViT-H) (Ilharco et al., 2021), an open-source implementation of CLIP. All backbone models are obtained from HuggingFace.

**Details of clustering recipe**  We explored several clustering strategies to better exploit the latent representations. Specifically, we considered three approaches: (1) clustering directly in the high-dimensional latent space, (2) applying dimensionality reduction followed by clustering, and (3) hierarchical clustering.

For clustering in the original latent space, we employed Gaussian Mixture Models (GMM) and K-means. For the dimensionality-reduction approach, we applied PCA and UMAP to reduce the latent representations to 32, 64, or 128 dimensions. Since performance was stable across these settings, we fixed the dimensionality to 64 in all subsequent experiments.

For hierarchical clustering, we first applied UMAP to reduce the latent dimension, followed by K-means clustering. In the first stage, the number of clusters was fixed at two. Each resulting cluster was then further subdivided by repeating the dimensionality reduction and clustering procedure. Across all methods, we varied the number of clusters over $\{2,4,8,10,16\}$, and we report the best Fowlkes–Mallows index achieved.

## B  DISCUSSION ON MULTI-CLASS FMI

In Section 3, we applied the Fowlkes–Mallows index (FMI) to evaluate both binary and multiclass labels, and observed clear numerical differences between the two cases. In this section, we aim to interpret the results for the multiclass labels and analyze what insights the numerical values provide.

To illustrate, we conduct a pseudo-label test. Specifically, we generate a random list $y_{true}$ with values ranging from 0 to 5. The length of $y_{true}$ is 5000. We then create predicted labels with probability $\alpha$, such that $P(y_{predict} = y_{true}) = \alpha$. We refer to $\alpha$ as the alignment level. We vary $\alpha$ across 20 linearly spaced levels and compute the FMI for each level. Since this is a random procedure, we repeat the process five times.

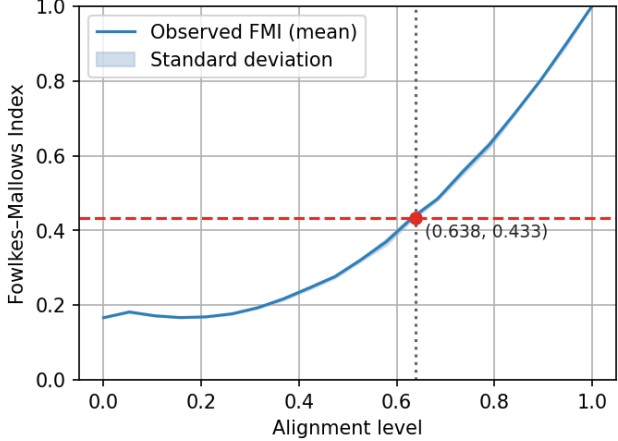

Figure 7: FMI Vs $\alpha$, where $\alpha$ denotes the agreement level with the ground-truth labels in the 6-class setting.

The results are shown in Figure 7. The narrow standard deviation region indicates low randomness. We highlight a point of 0.43 (CLIP result on Figure 1) on the curve. This corresponds to $\alpha = 0.64$, meaning that the predicted labels achieve approximately 64 % alignment with the ground-truth labels in the six-class setting. This result demonstrates that the CLIP model provides strong discriminative power for the race attribute.

## C  PROOF OF THEOREM 1

**Theorem 1** Given the demographic score function $\mathbf{s}_\theta(\mathbf{x}(t), t, d)$, after convergence:

$$\frac{d}{d\lambda}\mathrm{KL}(p_\lambda(x|d)||p(x|d)) = -(1-\lambda)\mathrm{Var}(l(x)) < 0.$$

Moreover, the within-cluster entropy:

$$H(p_\lambda(x|d)) \geq H(p(x|d)),$$

where $H(\cdot)$ denotes the entropy.

*Proof.* Take logarithm on the $p_\lambda(x|d)$:

$$\log p_\lambda(x|d) = \lambda \log p(x|d) + (1-\lambda)\log p(x) - A(\lambda), \tag{8}$$

We first compute the score function:

$$\nabla_x \log p_\lambda(x|d) = \lambda \nabla_x \log p(x|d) + (1-\lambda)\nabla_x \log p(x) \tag{9}$$

We omit $d$ for simplicity. Let $r_\lambda(x) = p(x|d)^\lambda p(x)^{1-\lambda}$, take the derivative:

$$\frac{\mathrm{d}}{\mathrm{d}\lambda}r_\lambda(x) = r_\lambda(x)\frac{\mathrm{d}}{\mathrm{d}\lambda}(\lambda \log p(x|d) + (1-\lambda)\log p(x)) = r_\lambda(x)l(x). \tag{10}$$

We define $A(\lambda) = \log Z(\lambda)$, therefore, we have:

$$Z(\lambda) = \int r_\lambda(x)dx, \tag{11}$$

Next, we compute the first derivative of $A(\lambda)$:

$$\begin{aligned} A'(\lambda) &= \frac{Z'(\lambda)}{Z(\lambda)} \\ &= \int \frac{r_\lambda(x)}{Z(\lambda)}l(x)\mathrm{d}x = \mathbb{E}_{p_\lambda(x|d)}[l] \end{aligned} \tag{12}$$

Then arbitrary integrable function $\phi$, to compute $\frac{\mathrm{d}}{\mathrm{d}\lambda}\mathbb{E}_{p_\lambda(x|d)}[\phi]$, using the quotient rules:

$$\begin{aligned} \frac{\mathrm{d}}{\mathrm{d}\lambda}\mathbb{E}_{p_\lambda(x|d)}[\phi] &= \frac{1}{Z}\int \phi\frac{\mathrm{d}r_\lambda(x)}{\mathrm{d}\lambda}\mathrm{d}x - \frac{Z'}{Z^2}\int \phi r_\lambda(x)\mathrm{d}x \\ &= \frac{1}{Z}\int \phi r_\lambda(x)l(x)\mathrm{d}x - \frac{1}{Z}\int r_\lambda(x)l(x)\mathrm{d}x\frac{1}{Z}\int \phi r_\lambda(x)\mathrm{d}x \\ &= \mathbb{E}_{p_\lambda(x|d)}[\phi l] - \mathbb{E}_{p_\lambda(x|d)}[\phi]\mathbb{E}_{p_\lambda(x|d)}[l] = \mathrm{Cov}(\phi, l) \end{aligned} \tag{13}$$

By definition:

$$\begin{aligned} \mathrm{KL}(p_\lambda(x|d)||p(x|d)) &= \int p_\lambda(x|d)(\log p_\lambda(x|d) - \log p(x|d)) \\ &= \mathbb{E}_{p_\lambda(x|d)}[\log p_\lambda(x|d) - \log p(x|d)] \end{aligned} \tag{14}$$

Using:

$$\log p_\lambda(x|d) = \lambda \log p(x|d) + (1-\lambda)\log p(x) - \log Z(\lambda) \tag{15}$$

We first deal with $\log p_\lambda(x|d) - \log p(x|d)$, substitute equation 15 into, we have

$$\log p_\lambda(x|d) - \log p(x|d) = (\lambda - 1)\log p(x|d) + (1 - \lambda)\log p(x) - \log Z(\lambda) \qquad (16)$$

We differentiate the KL divergence w.r.t $\lambda$

$$\frac{d}{d\lambda}\text{KL}(p_\lambda(x|d)||p(x|d)) = \mathbb{E}_{p_\lambda(x|d)}[\log p(x|d)] + (\lambda - 1)\frac{d}{d\lambda}\mathbb{E}_{p_\lambda(x|d)}[\log p(x|d)] -$$
$$\mathbb{E}_{p_\lambda(x|d)}[\log p(x)] + (1 - \lambda)\frac{d}{d\lambda}\mathbb{E}_{p_\lambda(x|d)}[\log p(x)] - A'(\lambda) \qquad (17)$$

We first deal with the second term $(\lambda - 1)\frac{d}{d\lambda}\mathbb{E}_{p_\lambda(x|d)}[\log p(x|d)]$, using equation 13

$$\frac{d}{d\lambda}\mathbb{E}_{p_\lambda(x|d)}[\log p(x|d)] = \text{Cov}(\log p(x|d), l) \qquad (18)$$

Because $A'(\lambda) = \mathbb{E}_{p_\lambda(x|d)}[l]$, thus

$$\mathbb{E}_{p_\lambda(x|d)}[\log p(x|d)] - \mathbb{E}_{p_\lambda(x|d)}[\log p(x)] - A'(\lambda) = 0. \qquad (19)$$

Therefore, the derivative of the KL w.r.t $\lambda$ can be simplfied as

$$\frac{d}{d\lambda}\text{KL}(p_\lambda(x|d)||p(x|d)) = (\lambda - 1)\text{Cov}(\log p(x|d), l) + (1 - \lambda)\text{Cov}(\log p(x), l) \qquad (20)$$
$$= (\lambda - 1)\text{Var}(\log p(x|d)) - (1 - \lambda)\text{Var}(\log p(x)) + (2 - 2\lambda)\text{Cov}(\log p(x|d), \log p(x))$$

We also have:

$$\text{Var}(l) = \text{Var}(\log p(x|d)) + \text{Var}(\log p(x)) - 2\text{Cov}(\log p(x|d), \log p(x)). \qquad (21)$$

Therefore

$$\frac{d}{d\lambda}\text{KL}(p_\lambda(x|d)||p(x|d)) = -(1 - \lambda)\text{Var}(l) < 0. \qquad (22)$$

We next discuss the entropy. By definition,

$$H(p_\lambda(x|d)) = -\lambda\mathbb{E}[\log p(x|d)] - (1 - \lambda)\mathbb{E}[\log p(x)] + A(\lambda), \qquad (23)$$

We differentiate twice w.r.t. $\lambda$, gives:

$$\frac{d^2}{d\lambda^2}H(p_\lambda(x|d)) = -\text{Var}(l) < 0, \qquad (24)$$

which is strictly concave in $\lambda$, which yields

$$H(p_\lambda(x|d)) > (1 - \lambda)H(p(x)) + \lambda H(p(x|d)) \qquad (25)$$

We assume that $H(p(x)) > H(p(x|d))$, which holds because the diversity of the unconditional distribution is greater than that of the conditional distribution.

Therefore, we have

$$H(p_\lambda(x|d)) > H(p(x|d)) \qquad (26)$$

$\square$

## D  FAIR SAMPLING ALGORITHM

We present our fair sampling algorithm in Algorithm 2.

---

**Algorithm 1** Demographics identification

---

**Input:** Gaussian noise $\mathbf{x}_0$, Denoising timesteps $T$, Dimension reduction function UMP, Initial Searching range $N$, Image backbone $f(\cdot)$, The number of clusters $c, v$.
**Output:** Demographics clusters $\mathbf{C}$.
**for** $i = 1, ..., N$ **do**
    Generate $\mathbf{x}_T^i$ with $\mathbf{x}_0$.
    $\mathbf{z}^i = f(\mathbf{x}_T^i)$                                 *Latent representation*
**end for**
$\mathbf{Z} = \texttt{Concatenate}(\mathbf{z}^i)$
$\mathbf{Z}_{\text{reduced}} = \text{UMAP}(\mathbf{Z})$
$\{\mathbf{c}_i\}_{i=1}^c = k\text{means}(\mathbf{Z}_{\text{reduced}})$                           *Coarse clustering*
**for** $i = 1, ..., c$ **do**
    $\mathbf{Z}_{\text{red}}[c_i] = \text{UMAP}(\mathbf{Z}[c_i])$
    $\{\mathbf{c}_{i,j}\}_{j=1}^v = k\text{means}(\mathbf{Z}_{\text{red}}[c_i])$                   *Fine-grid Clustering*
**end for**
$\mathbf{C} = \bigcup_{i=1,...,c, j=1,...,v} \mathbf{c}_{i,j}$                             *Form clusters*
**return** $\mathbf{C}$

---

**Algorithm 2** Fair sampling generation

---

**Input:** Denoising timesteps $T$, Total generate image number $N$, Guidance strength $\lambda$.
**Output:** Generated samples $\mathbf{x}_T$.
[1]
**for** $i = 1, ..., |\mathbf{C}|$ **do**
    **for** $t = 1, ..., T$ **do**
        $\mathbf{s}_\theta(\mathbf{x}(t), t | i) = \mathbb{E}_{\mathbf{x}(t)}[\mathbf{s}_\theta(\mathbf{x}(t), t | \mathbf{x}(T) \in \mathbf{C}_i)]$     *Demographic representative score function.*
        $\mathbf{s}_\theta^\perp(\mathbf{x}(t), t, i) = \mathbf{s}_\theta(\mathbf{x}(t), t | i) - \frac{\langle \mathbf{s}_\theta(\mathbf{x}(t), t | i), \mathbf{s}_\theta(\mathbf{x}(t), t) \rangle}{\langle \mathbf{s}_\theta(\mathbf{x}(t), t), \mathbf{s}_\theta(\mathbf{x}(t), t) \rangle} \mathbf{s}_\theta(\mathbf{x}(t), t)$.
    **end for**
**end for**
$n = \frac{N}{|\mathbf{C}|}$
**for** $c = 1, .., |\mathbf{C}|$ **do**
    **for** $i = 1, ..., n$ **do**
        $\mathbf{s}_\theta(\mathbf{x}(t), t, c) = \lambda \mathbf{s}_\theta^\perp(\mathbf{x}(t), t, c) + \mathbf{s}_\theta(\mathbf{x}(t), t)$.
        **for** $t = 0, ..., T - 1$ **do**
            $\mathbf{x}_{t+1} = \frac{1}{\sqrt{\alpha_t}}(\mathbf{x}_t - \frac{1-\alpha_t}{2\sqrt{1-\bar{\alpha}_t}} \mathbf{s}_\theta(\mathbf{x}(t), t, c))$     Sampling with the stable demographic score
function (ODE sampling example)
        **end for**
    **end for**
**end for**
**return** $\mathbf{x}_T$

---

## E EXPERIMENT DETAILS

Our experiments are conducted on Nvidia A5000 GPUs, each equipped with 24 GB of GPU memory, using CUDA version 12.2.

**Details for section 4.1** For the CelebA dataset, we employ the pretrained weights for the UNet model from Hugging Face, using the 'google/ddpm-celebahq-256' configuration. The generated image resolution is $256 \times 256$. We set the initial number of images to 600. We set $\lambda = 0.07$, $c = 2$, $v = 3$, DDIM scheduler, and the number of inference steps to 50. The total runtime for our algorithm is 3 hours using a single A5000 GPU.

**Details for section 4.2** For Stable Diffusion V2.1 and Stable Diffusion V3, the image generation resolution is $512 \times 512$. We set the initial image generation for exploring bias behavior to 600. The image encoder used is a ViT-H-14, pretrained with the dataset "laion2b_s32b_b79k". We have adapted the clustering algorithm from (Sohoni et al., 2020). During the fair generation stage, the $\lambda$ is set to 0.3. We use "DPMSolver Single step Scheduler" and set the number of inference steps to 28. The total run time for our algorithm with the DPM solver is 1 hour to generate 200 fairer samples.

**Details for section 4.3** We train a ResNet-18 model using the SimCLR algorithm. For this, we employ the Adam optimizer with a learning rate of $3 \times 10^{-4}$, a weight decay of $10^{-4}$, a batch size of 256, and we train for 256 epochs. To ensure a fair comparison, each dataset is sized at 1000 samples. To evaluate the representations learned, we employ a linear probe on the CelebA dataset. The entire training set is used to train the linear classifier, utilizing the Adam optimizer with a learning rate of $3 \times 10^{-4}$ and no weight decay, and a batch size of 128.

**Clusters visualization** We present visualizations of sample clusters, randomly selecting four images per cluster as shown in Figure 9. The images within each cluster demonstrate shared characteristics.

# F    EVALUATION ON CLUSTERING QUALITY

## F.1    FOWLKES-MALLOW SCORE

A commonly used evaluation metric in clustering research is the silhouette score. However, the silhouette score is limited by its inability to incorporate labels into its evaluation. To address this limitation, we employ the Fowlkes-Mallows score, which measures the similarity between two sets of clusterings. The Fowlkes-Mallows score (Fowlkes & Mallows, 1983) is computed as the geometric mean of pairwise precision and recall between the clustering results and the ground truth labels:

$$FM = \sqrt{\frac{TP}{TP + FP} \frac{TP}{TP + FN}} \tag{27}$$

where $TP$, $FP$, and $FN$ represent true positives, false positives, and false negatives, respectively, utilizing the ground truth labels of the dataset for computation.

## F.2    EVALUATION PROCEDURE IN SECTION 3.1

The evaluation assesses how effectively the clustering algorithm aligns with demographic information. Directly utilizing the Fowlkes-Mallows score can be problematic in scenarios of over-clustering, where more clusters are generated than the number of labeled categories. This results in multiple clusters correlating with a single label. When applied directly, the Fowlkes-Mallows score considers only one cluster associated with a specific ground truth label, disregarding any other clusters related to that label.

To address this issue, we introduce a mapping function that identifies the most frequent label within each cluster. This approach enables more accurate evaluation of how well clusters represent their respective labels, particularly in scenarios of over-clustering. Define the set of indices:

$$I_c = \{1, 2, ..., |\mathbf{C}|\} \tag{28}$$

For each cluster $\mathbf{c}$ and each label $y \in \mathcal{Y}$, define the count:

$$n_{\mathbf{c},y} = \sum_{i \in I_c} \delta_{y_i, y}, \tag{29}$$

where

$$\delta_{y_i, y} = \begin{cases} 1 & \text{if } y_i = y \\ 0 & \text{if } y_i \neq y \end{cases} \tag{30}$$

Then, determine the most common label for each cluster:

$$y_{\mathbf{c}}^* = \arg \max_{y \in \mathcal{Y}} n_{\mathbf{c},y}. \tag{31}$$

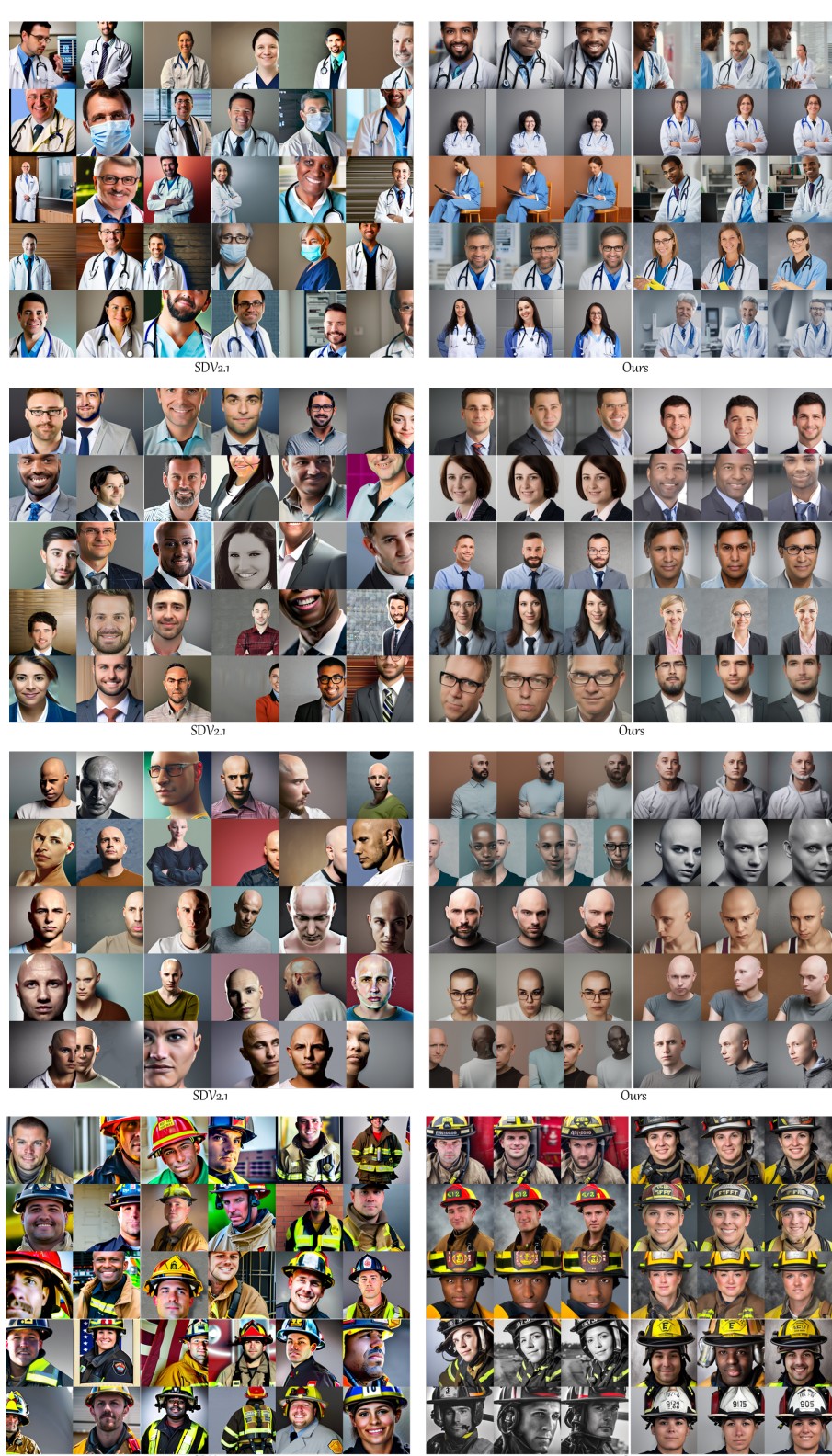

Figure 8: Image quality evaluation. Left: SDV2-generated images corresponding to the prompts Doctor, Manager, Bald, and Firefighter. Right: clustering results, where we visualize 3 randomly selected samples from each cluster, with 10 clusters per occupation or appearance.

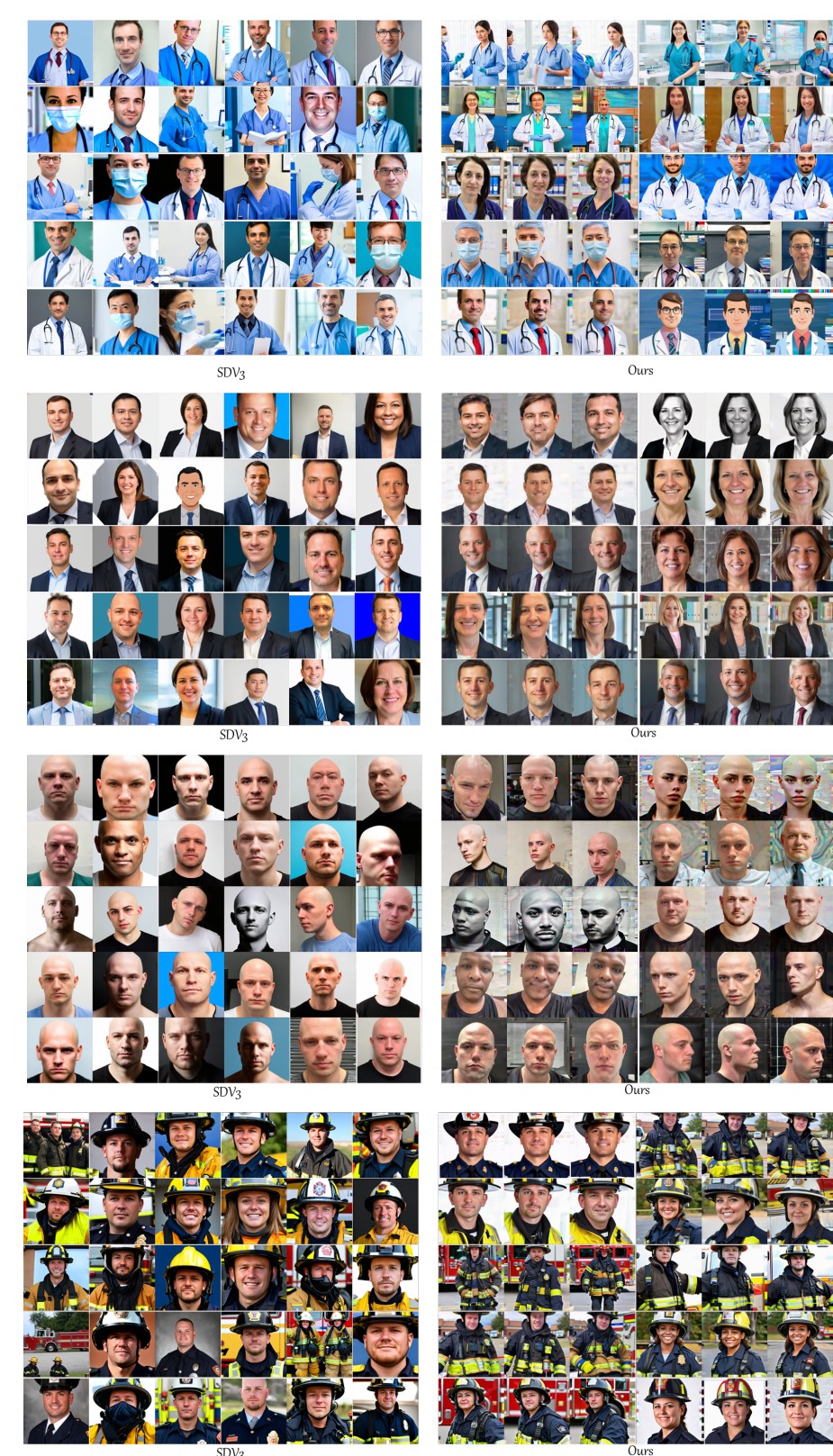

Figure 9: Image quality evaluation. Left: SDV3-generated images corresponding to the prompts Doctor, Manager, Bald, and Firefighter. Right: clustering results, where we visualize 3 randomly selected samples from each cluster, with 10 clusters per occupation or appearance.

The mapping function $g$ from cluster to label is then:

$$g(\mathbf{c}) = y_{\mathbf{c}}^*. \tag{32}$$

We utilize mapped clusters and annotated demographic information from CelebA and FairFace to compute the Fowlkes-Mallows score in section 3.1.

## G    DISCUSSION OF EVALUATION METHODS FOR GENERATED IMAGES

For evaluation purposes, we assign pseudo-demographic attributes using classifiers. This section discusses the rationale behind the selection of these classifiers and the accuracy they bring to the evaluation process.

**Text-to-image evaluation.**    As text-to-image models are trained on extensive datasets comprising diverse text-image pairs, the resulting images exhibit varied styles and patterns. Consequently, using a supervised classifier is not advisable due to the high risk of distribution shift between the training set and the generated images. Therefore, we utilize zero-shot classification in vision-language models (VLMs) as the classifier to assign pseudo-demographic attributes. We first evaluate the performance of vision-language models on the task of identifying race in a multi-class classification setup. Our methods are assessed using the FairFace dataset, which includes race annotations. The VLMs model we select from CLIP-ViT-H-14 (Clip-H) (Ilharco et al., 2021), CLIP-VIT-L/14 (CLIP-L) (Ilharco et al., 2021), XVLM-2 (Zeng et al., 2023). All comparison backbone models share the same text prompt ['A figure of an Asian person.', 'A figure of an Indian person.', 'A figure of A Black person.', 'A figure of A White person.', 'A figure of A Middle East person.', 'A figure of A Latino Hispanic person.']. The results are shown in Table 6. We observe that CLIP-H achieves the best performance in race identification. We extend our evaluation of CLIP-H to gender identification using the FairFace and CelebA datasets. The results are presented in Table 7. The gender text prompts is ['A figure of a male.', 'A figure of a female.']. We observe that CLIP-H achieves high accuracy on both the FairFace and CelebA datasets, indicating its robustness and suitable for the evaluation purpose.

Table 6: Top-1 accuracy for zero-shot classification with race using different backbone models

| Methods | Clip-H | Clip-L | XVLM-2 |
|---------|--------|--------|--------|
| Acc     | 73.31  | 72.41  | 24.78  |

Table 7: Top-1 accuracy for zero-shot classification with gender using CLIP-H model

|     | Gender (FairFace) | Gender (CelebA) |
|-----|-------------------|-----------------|
| Acc | 96.20             | 98.18           |

**Intersectional bias evaluation**    We explore intersectional bias concerning two binary attributes: $Gender$ and $Age$. In our text-to-image model, we utilize zero-shot classification capabilities from the CLIP model. To quantify the evaluation more effectively, we conduct experiments on the CelebA test set, annotated with 'male' and 'young'. We employ the CLIP-ViT/L-14 for zero-shot classification using text prompts such as ['A figure of a male.', 'A figure of a female.'], ['A figure of a not young person.', 'A figure of a young person.']. We assess these prompts across all demographic group combinations to measure group-specific performance accuracy. The results are presented in Table 8.

**Unconditional image generation evaluation**    For unconditional image generation, we employ pre-trained weights from the CelebA dataset and further train a denoising network for the FairFace dataset. The images generated by the model should align with the distribution of the original dataset. Therefore, for attributes that a dataset annotates, such as gender and age, we apply a supervised

Table 8: Intersectional bias evaluation CLIP-ViT/L-14

|     | Gender = 0, Age = 0 | Gender = 0, Age = 1 | Gender = 1, Age = 0 | Gender = 1, Age = 1 | Overall |
|-----|---------------------|---------------------|---------------------|---------------------|---------|
| Acc | 67.22               | 72.99               | 63.40               | 70.73               | 70.49   |

Table 9: Unconditional diffusion models classifiers accuracy

| Methods            | Gender | Age   |
|--------------------|--------|-------|
| DINO-V2 (CelebA)   | 98.42  | 88.72 |
| DINO-V2 (FairFace) | 90.06  | 86.64 |

method to assign pseudo-demographic labels. For attributes not annotated in a dataset, such as race, we apply the same pseudo-labeling method introduced for text-to-image evaluation. We use DINO-V2 (Oquab et al., 2023) as the image encoder and train a linear classifier with the training set, the results are shown in Table 9. We observe that age classification underperforms compared to gender classification due to the binary labeling of age. This binary approach to age annotation introduces subjectivity, which contributes to the observed performance disparity between gender and age classifications. Nevertheless, the DINO-V2 backbone maintains strong performance across various datasets with differing annotations. Accordingly, we utilize DINO-V2 and apply a linear classifier trained on the specific annotations for each dataset. For the intersectional bias evaluation of the unconditional models, we employ the same linear classifier using DINO-V2 representations to ensure robust and accurate assessments. We predict each generated sample and categorize these samples into groups based on combinations of demographic attributes.

## H  BASELINE METHODS

In this section, we introduce the comparison methods related to the fair diffusion model in the sampling stage.

- Attribute Switching (AS) (Choi et al., 2024): Attribution Switching is a method that controls the generation path via conditional inputs. Specifically, during the generation step, this method identifies a transition point. For timesteps before the transition point, the denoising network processes one attribute (e.g., male). For timesteps after the transition point, it switches to the alternate attribute (e.g., female). This approach controls the final generation's attribution without the need for a classifier at each timestep. However, it requires modifying the network structure to incorporate the attribution as an input, which means that publicly available pre-trained weights cannot be utilized due to these structural changes.

- Latent Editing (LE) (Kwon et al., 2023): Latent editing is a technique that modifies the generation process to achieve semantic changes in the final results. Specifically, in the initial stage, the latent space (H-space) of the UNet is guided by a convolutional neural network (CNN) classifier, which is trained using representations from the H-space to identify desired attributes. This setup enables the editing of various demographic attributes based on the CNN classifier, which is trained on distinct annotations. For fair generation, Latent Editing (LE) combines images with different edited annotations to facilitate a fairness-aware sampling process.

- H-guidance (Parihar et al., 2024): This work builds on the findings of Kwon et al. (2023), which demonstrated that direct image editing results in suboptimal image generation quality, a conclusion that aligns with our observations. Building upon this insight, H-guidance is employed as another classifier-guided method to generate specific attributes. For fairness considerations, H-guidance replaces manual adjustments in image editing strength with a loss function that determines the necessary edits to achieve demographic balance. Moreover, this approach simplifies the classifier to a series of linear classifiers.

## I   MORE ABLATION STUDY

**The number of clusters**   We investigate the effect of the number of clusters on fairness outcomes. Specifically, we vary $v = \{2, 3, 4, 5, 6, 8\}$, which corresponds to cluster sizes of $|\mathbf{C}| = \{4, 6, 8, 10, 12, 16\}$. All other parameters are kept fixed, and images are generated under each setting. The results are shown in Figure 10.

We observe that Gender remains relatively stable across all values of $v$. This is because our hierarchical clustering algorithm fixes $c = 2$, and our findings confirm that gender information is more prominent and easily distinguishable, leading to stable fairness outcomes.

For Age, smaller cluster numbers yield better results. For Race, too few clusters prevent effective recognition of multiple racial groups, resulting in poor performance. Conversely, too many clusters lead to over-clustering, which also reduces performance. Based on these observations, we recommend $k = 3, 4, 5$ as a reasonable choice that balances fairness across different attributes.

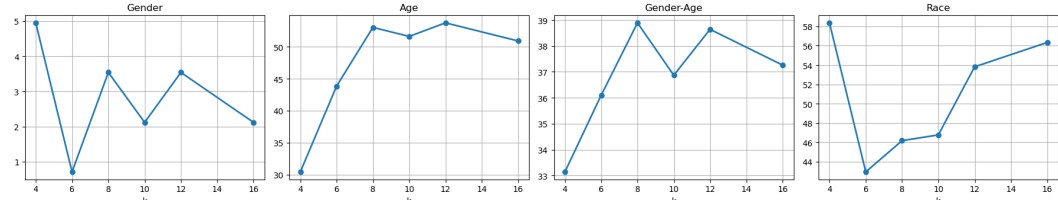

Figure 10: Ablation study on the number of clusters $|\mathbf{C}|$.

## J   FAIRNESS METRICS IN CLASSIFICATION TASKS

We employ the concept of group fairness in evaluating classification tasks. Specifically, we concentrate on three metrics: Demographic Parity (DP), Equal Opportunity (EOp), and Equalized Odds (EOd). DP ensures that the predictive rates are equivalent across diverse demographics. EOp mandates that the true positive rates are consistent, while EOd requires equality in both true positive and false positive rates. These metrics are formally computed as follows (Lu et al., 2024):

$$
\begin{aligned}
\text{DP} &= |PR_i - PR_j|, \\
\text{EOp} &= |TPR_i - TPR_j|, \\
\text{EOd} &= \frac{1}{2}(|TPR_i - TPR_j| + |FPR_i - FPR_j|), \ \ i, j \in \mathcal{A}.
\end{aligned}
\tag{33}
$$

In the equation, $PP, TPR, FPR$ denote the predictive rate, true positive rate, and false positive rate, respectively.

## K   LIMITATION DISCUSSION

Despite the strong performance of our proposed method, our approach to addressing the fair diffusion problem follows a two-step philosophy: first, identifying underrepresented groups, and then promoting their representation. This approach acts as a remedial mechanism for mitigating biases inherent in pre-trained models. However, if certain demographic combinations are entirely absent in the initial identification stage, our algorithm fails to detect them, limiting its effectiveness. Addressing this limitation remains an important direction for future research.

## L   THE USE OF LARGE LANGUAGE MODELS (LLMs)

We employ ChatGPT-5 to improve sentence-level readability in the paper. Each modified sentence is carefully checked by the authors to ensure that the meaning and correctness remain unchanged.

We also use Cursor to convert author-written notebook code into scripts for parallel computing for large-scale evaluation. For the literature review, we combine human search with ChatGPT-5, where the model is used only for retrieval purposes. We do not rely on ChatGPT to summarize the referenced papers.

