# OpenReview forum: "Fair Diffusion Sampling without Demographics"
_ICLR.cc/2026/Conference — ICLR 2026 Conference Withdrawn Submission_

### Official Review · Reviewer_gaAm · 2025-10-21

**Soundness:** 2
**Presentation:** 1
**Contribution:** 2
**Rating:** 2
**Confidence:** 5

**Summary:**

This paper introduces an approach to both detect and control it at sampling time without predefined labels. It treats social bias as embedded in pre-trained image encoders (e.g., CLIP, DINO), uses those representations to uncover demographic structure via clustering, and then applies a demographic score to steer generation toward underrepresented groups. This approach aims to handle multiple and intersectional biases across both conditional and unconditional models without prior demographic labels.

**Strengths:**

The paper tackles a timely problem of detecting bias and controlling generation without expensive retraining, unlike approaches that mitigate only known biases.

**Weaknesses:**

1. I found the paper hard to follow because the information is scattered. For example, key details about the clustering appear in the appendix (around line 162), but the main text doesn’t properly reference that section.
2. The main table is also difficult to parse. There are so many combinations that it’s unclear what the reader should focus on. In addition, the figure and table captions are vague and don’t clarify the content.
3. The central motivation is to discover demographic attributes without preset labels. I agree that testing with known labels matters, but the qualitative results only reflect known biases. The paper offers no evidence that the method finds new, previously unknown demographic biases, which is surprising.
4. The qualitative results raise concerns. Despite the claim that Figure 3 shows three random samples from each demographic cluster, the images are near duplicates, implying a lack of diversity. The Stable Diffusion examples in Figures 8 and 9 exhibit the same problem.
4. Fairness research often uses CLIP to assess gender and racial bias, with studies such as https://arxiv.org/abs/2311.17216 indicating that certain race classifications are not reliable. Because this approach clusters directly on CLIP embeddings, it may not cluster races well, which is also kind of reflected in the poor race-related performance in Table 2.
5. Image encoders themselves can be biased, and this approach risks introducing an additional layer of bias into the pipeline.

**Questions:**

1. Can you provide some qualitative results for clusters corresponding to different races and some intersectional demographic attributes?

---

### Official Review · Reviewer_3Vss · 2025-10-23

**Soundness:** 2
**Presentation:** 3
**Contribution:** 2
**Rating:** 4
**Confidence:** 3

**Summary:**

This paper addresses the fairness issues in the diffusion models. Specifically, it hypothesises that biases in generation are a characteristic embedded within pre-trained image encoders. Then, it proposed identifying bias elements in the widely used encoder backbones. Finally, fair diffusion processing is achieved via encouraging a more uniform distribution across identified bias groups.

The proposed method is technically sound, but I have some concerns about how the clusters align with the actual underlying attribute. I request the author to elaborate on this point, and I am willing to revise the rating.

**Strengths:**

- This paper proposes to uncover the demographic structure of diffusion models by analysing the image encoder. Assuming that image encoders are trained on large-scale real-world images, they inevitably encode social biases. Hence, it proposes to extract demographic information through a hierarchical clustering algorithm.
- This paper defines a demographic score function (eqn 6) to promote the representation of underrepresented groups. Empirical evidence shows that this work is capable of mitigating multiple sources of bias across both conditional and unconditional diffusion models.
- The implementation details are clearly stated, and the code is available together with the submission.

**Weaknesses:**

The underlying hypothesis of this work is that features extracted from the (foundation) image encoder have coupled information related to the (sensitive) attribute and other class-based information. Through a clustering approach, it is possible to identify clusters that share common properties. In reality, it is not evident how the success in identifying such clusters may be related to sensitive attributes and contribute to fair generation. For example, the clusters may characterise image style or human pose, rather than the desired target attribute.

In addition, a clustering algorithm may not guarantee that the underlying cluster can cover the targeted demographic. Hence, the selection of the number of clusters has a strong implication for the model performance in fair image generation.

**Questions:**

- Could the author explain the concerns about the sensitive data collection?
- Can bias be mitigated without prior knowledge of demographics? I am sure that the proposed method does not require human annotation; however, it still necessitates knowledge of the factors typically considered in bias mitigation (e.g., gender, age, race, etc.). This knowledge itself is undoubtedly required to determine the number of clusters, or to determine a good c and v in Section 3.
- In Sec 3.1 (Fig 2) and Sec 4.1 (Tab 2), the age is a binary attribute. Is this a typical setup in the literature? Could the author provide more details on what the underlying label (young vs. old) is?
- In Fig. 8 and 9, each cluster has produced images that share very similar properties (e.g., underlying attribute, image composition, and style). Hence, the diversity may be highly related to the number of clusters. I am curious, does it really contribute to fair image generations?
- For Table 4, it would be beneficial to show more results with different classification tasks (i.e., target label) and sensitive attributes.
- What is the performance of the DINO-V2-based linear classifier for controlled images?

Suggestions:
- Please use consistent colour labels across all four attributes; it may improve the readability and comparison across each attribute.
- There are minor errors in citation references in the main text.

---

### Official Review · Reviewer_sWNm · 2025-10-31

**Soundness:** 1
**Presentation:** 2
**Contribution:** 2
**Rating:** 2
**Confidence:** 5

**Summary:**

This paper proposes a novel method for mitigating bias in diffusion models at the sampling stage, with the primary contribution being that it does not require explicit demographic annotations. The core idea is to leverage the fact that pre-trained image encoders (like CLIP) already embed social biases.

The method consists of two main stages:
1. Demographic Group Identification: The authors use a pre-trained encoder to extract latent features and then apply a two-stage hierarchical clustering algorithm to automatically discover underlying demographic groups.
2. Controlled Fair Generation: A novel "stable demographic score" function is introduced. This function guides the sampling process to promote a more uniform distribution across the discovered clusters, while using an orthogonalization technique to maintain sample diversity and quality.

Experiments are conducted on unconditional generation, text-to-image stereotype mitigation, and a downstream classification task to demonstrate the method's effectiveness.

**Strengths:**

1. The paper tackles the critical and highly practical challenge of mitigating bias in generative models without relying on sensitive, costly, and privacy-invasive demographic labels.

2. The core conceptual idea of "weaponizing" the inherent biases of pre-trained encoders to perform annotation-free debiasing is highly of significant interest.

**Weaknesses:**

1. The paper's entire premise rests on a fatal, unproven assumption: that unsupervised clustering of the encoder's latent space will always find sensitive attributes. For example, when setting $c=2$, the paper provides no guarantee that the clusters represent 'male/female' instead of 'long hair/short hair' or 'smiling/not smiling'. The paper validates this assumption using labels (Fig 1), but then applies it (Eq 3) in a setting where no such validation is possible. This is not 'in the wild'; it's a leap of faith that renders the method scientifically unsound.
2. The claim of being "annotation-free" is directly contradicted by the use of oracle hyperparameters. The choice of $c=2$ is explicitly justified by its prior, known correlation with gender. This is not unsupervised discovery; it's tuning the method to a specific dataset and bias, which is the exact opposite of the "in the wild" claim.
3. The method is not a clear improvement, as it introduces new problems. In the T2I experiments (Table 3), the method worsens racial fairness for SDV2.1 (Avg R FD 49.89 vs. 51.96). This suggests the method is not a general debiaser but a blunt instrument that may trade one bias for another.
4. The method results in a worse (higher) FID score than the vanilla DDIM baseline (78.11 vs. 71.50), indicating a loss of generative quality.

**Questions:**

See weaknesses above

---

### Note · Authors · 2025-11-20

I have read and agree with the venue's withdrawal policy on behalf of myself and my co-authors.